# Effect of the Presence of Lignin from Woodflour on the Compostability of PHA-Based Biocomposites: Disintegration, Biodegradation and Microbial Dynamics

**DOI:** 10.3390/polym15112481

**Published:** 2023-05-27

**Authors:** Patricia Feijoo, Anna Marín, Kerly Samaniego-Aguilar, Estefanía Sánchez-Safont, José M. Lagarón, José Gámez-Pérez, Luis Cabedo

**Affiliations:** 1Polymers and Advanced Materials Group (PIMA), Universitat Jaume I (UJI), Avenida de Vicent Sos Baynat s/n, 12071 Castelló de la Plana, Spain; pfeijoo@uji.es (P.F.); anmarin@uji.es (A.M.); samanieg@uji.es (K.S.-A.); esafont@uji.es (E.S.-S.); gamez@uji.es (J.G.-P.); 2Novel Materials and Nanotechnology Group, Institute of Agrochemistry and Food Technology (IATA), Spanish National Research Council (CSIC), Calle Catedrático Agustín Escardino Benlloch 7, 46980 Paterna, Spain; lagaron@iata.csic.es

**Keywords:** PHBV, woodflour, sustainable bioplastics, aerobic composting, biodegradation, microbial population

## Abstract

Poly(3-hydroxybutyrate-*co*-3-hydroxyvalerate) (PHBV) has gained attention as a possible substitute for conventional polymers that could be integrated into the organic recycling system. Biocomposites with 15% of pure cellulose (TC) and woodflour (WF) were prepared to analyze the role of lignin on their compostability (58 °C) by tracking the mass loss, CO_2_ evolution, and the microbial population. Realistic dimensions for typical plastic products (400 µm films), as well as their service performance (thermal stability, rheology), were taken into account in this hybrid study. WF showed lower adhesion with the polymer than TC and favored PHBV thermal degradation during processing, also affecting its rheological behavior. Although all materials disintegrated in 45 days and mineralized in less than 60 days, lignin from woodflour was found to slow down the bioassimilation of PHBV/WF by limiting the access of enzymes and water to easier degradable cellulose and polymer matrix. According to the highest and the lowest weight loss rates, TC incorporation allowed for higher mesophilic bacterial and fungal counts, while WF seemed to hinder fungal growth. At the initial steps, fungi and yeasts seem to be key factors in facilitating the later metabolization of the materials by bacteria.

## 1. Introduction

Long-lasting plastic pollution is globally ubiquitous. As a consequence of the mismanaged waste collection system and the increasing single-use throw-away culture, 79% of the global plastic waste generated has ended up in the environment [1]. Actually, 40% of plastic packaging produced comprises more than 50% of plastic-waste share because they are produced and discarded in the same year [2]. Furthermore, the contamination of both plastic and organic residues by each other hinders their respective management. Attention has been focused on biopolymers as an eco-friendly solution to replace conventional plastics and, thus, simplify and optimize the treatment of plastics together with urban waste by organic recycling (also known as composting) [3,4]. In this sense, both biodegradable and biobased polymers would allow the transformation from a linear economy into a biocircular system [5].

Among other biopolymers, PHBV is one of the most promising polyhydroxyalkanoates (PHA) due to its commercial availability and mechanical performance similar to polypropylene (PP) [6]. Additionally, PHBV is a bacterial polyester potentially produced from sustainable feedstock [7,8] and biodegradable in all ecosystems (soil, industrial compost, home compost, marine, and anaerobic environments) [9,10,11]. However, its high intrinsic fragility and narrow processing window hinder its applicability [5]. Its presence in the market is also limited by its expensive cost of production: over 5.5 US$/kg compared to 1.2 US$/kg for conventional polymers [12]. Therefore, direct replacement is not yet a realistic scenario.

Incorporation of fillers into polymers, thus obtaining a composite, is a common strategy to reinforce materials and modulate their cost by using residues [13]. Wood is one of the most commonly used fillers [14], and it is chemically composed of three main polymers (see Appendix A [15,16]): (i) cellulose, a linear polymer of glucose units; (ii) hemicellulose, a mixture of highly branched polysaccharides; iii) lignin, a complex three-dimensional (3D) polymer that contains aromatic hydroxyl groups [15]. The current and extensive literature about wood-PHA composites is focused on improving mechanical performance or filler-matrix adhesion by processing or chemical purification to remove lignin and waxes [17,18,19]. Although all these research studies highlight the biodegradability of the so-obtained composites, only a few considered studying it (see Appendix A in Appendix A) [5,18,19,20,21,22,23,24,25,26,27]. For instance, Avella et al. [23] found a lower disintegration rate from 20% of fiber content in PHB-wheat straw composites under composting conditions. It was attributed to the non-favorable thermophilic conditions for lignin fungal degraders. By contrast, our group [5] reported no remarkable differences in weight loss regarding the type and content (up to 20%) of lignocellulosic residue after 30 days of composting. Literature about the biodegradation of PHA-lignocellulose composites in compost is mainly focused on weight loss and visual appearance changes, and although such observations are useful, there is a lack of knowledge in terms of ultimate biodegradability and mechanisms. To the best of our knowledge, only Chan et al. [24] have reported a more comprehensive study about the biodegradation process of woodflour-PHBV considering changes over time of micro-structure, weight loss, mechanical properties, and oxygen consumption (bioassimilation). However, the environment studied was soil. A comparison table between this study and the literature found in terms of characterization and biodegradation of PHA-based composites in compost and soil has been included in the Appendix A (see Appendix A) [5,18,19,20,21,22,23,24,25,26,27].

When designing a new material/product, it is necessary to take into account not only its target application, properties, and cost but also its whole life cycle in terms of origin, reusability, recyclability, and eventual end-of-life [28]. The natural end-of-life of organic matter in nature is biodegradation. Biodegradability is the susceptibility of materials to be biodegraded in specific conditions, while biodegradation is the process by which microorganisms and their enzymes break down materials into small molecules and subsequently assimilate them [29]. In the same way, compostability is the ability of a product to be organically recycled as organic matter during a conventional composting cycle (i.e., achieve full biodegradation under composting conditions during a convenient period of time) [30]. The susceptibility of a material or a product to undergo biodegradation in a specific condition depends on abiotic/environmental factors (temperature, moisture, pH, oxygen, etc.), biotic factors (type and proportion of microbial communities, enzymes, etc.), physicochemical properties of the polymer and components of the composite (chemical structure, melting and glass transition temperature, crystallinity, molecular weight, hydrophilic/hydrophobic properties, etc.), and shape and size of the particular product (thickness, surface area, roughness, etc.) [28,30,31].

Since organic recycling is currently an established waste management system in Europe and North America, the industrial (or thermophilic) composting environment is one of the most important [30]. The increasing variability of bioplastics (including composites) encourages composting facilities to implement innovative organic waste treatments, and, as mentioned previously, it makes necessary deeper research into compostability [30,32]. The wide presence of lignocellulosic fillers in the biopolymer industry highlights the importance of understanding their impact on biodegradability either in a synergistic or antagonistic way. Moreover, their incorporation brings on further uncertainty about the waste management and environmental impact of these composite materials [33].

The aim of this work was to investigate the impact of lignocellulosic fibers on the biodegradability of PHBV/woodflour composite under thermophilic composting conditions while considering its service performance as a realistic biomaterial alternative. This comprehensive study encompasses conventional biodegradation studies, such as disintegration and mineralization, as well as microbiological aspects to unveil the natural process that governs the ultimate end-of-life of biodegradable polymers for short-life plastic applications. This hybrid approach provides a more complete understanding of the properties and environmental impact of PHA/woodflour composites, filling a critical gap in the current body of knowledge between biodegradation and materials science, as evidenced by the lack of literature references that include analysis of all fields mentioned.

## 2. Materials and Methods

### 2.1. Materials

ENMAT Y1000P PHBV was purchased from Tianan Biologic Material Co. (Ningbo, China) in pellet form (3 wt.% hydroxyvalerate, density 1.23 g/cm^3^). Pine woodflour (WF) was procured by J. Adrian S.L. (Burjassot, Spain). Two grades of purified alpha-cellulose were acquired from CreaFill Fibers Corp. (Chestertown, MD, USA): TC90 (cellulose > 99%) for composites and TC40 (cellulose > 99.6%) as reference for the biodegradation test (positive control). All culture media for microbiological characterization were provided by Laboratorios Conda S.A (Madrid, Spain). Antibiotic and antifungal compounds were acquired from Merck Life Science S.L. (Madrid, Spain). Solvent 2,2,2-trifluoroethanol (TFE) was purchased from Sigma-Aldrich, and glycerol from Scharlab S.L. (Barcelona, Spain). Vegetal mature compost with a C/N ratio of 12.1, moisture of 32%, and a pH value of 8.5 was supplied by Hermanos Aguado S.L. (Toledo, Spain).

### 2.2. Preparation of Fiber/PHBV Composites

Prior to extrusion, PHBV and fibers (TC90, WF) were dried at 60 °C for 24 h in a Piovan DPA 10 dehumidifier (Santa Maria di Sala, Italy). Composites were prepared in a Haisin TSE-20B (Nanjing, Jiangsu, China) co-rotating twin-screw extruder (Ø = 22 mm, L/D ratio = 32–44). The rotation speed was 200 rpm, and the temperature profile was set at 175/173/165/165/165/165/165 °C from hopper to nozzle. All the components were manually dry-mixed. The extruded material was cooled in a water bath and pelletized. Pellets of each formulation were directly used for rheological measures. For the biodegradation test, the powder of the pellets was obtained by cryogenic milling under liquid nitrogen and subsequent sieving through a 250 µm mesh. For SEM, TGA and disintegration, films of 400 µm thickness were obtained by compression molding using a hot-plate Carver press (Wabash, IN, USA) at 180 °C and 4 bar for 4 min. Samples are named PHBV, PHBV/TC, and PHBV/WF, indicating the type of fiber, having, in all cases, 15 wt.% fiber content. The fiber content in the composites was selected based on the average value of the most studied fiber contents (10, 20%) in literature [14,23,31].

### 2.3. Characterization of Fibers and Composites

The morphology of the fibers and the polymeric materials was studied by SEM using a high-resolution field-emission JEOL 7001F microscope (Peabody, MA, USA) at a voltage of 5 kV. Prior to observation, the films of the materials were fractured in liquid nitrogen to avoid plastic deformation and coated by sputtering with a thin layer of platinum. The average particle size and aspect ratio (L/D) of the fibers were calculated from the SEM micrographs.

The moisture and ash content of fibers were determined according to the TAPPI standards T210 and T211, respectively. Samples were previously conditioned at 23 °C and 50% relative humidity (% R.H.) until the equilibrium was reached.

Wide-angle X-ray scattering (WAXS) experiments were conducted on fibers using a Bruker AXS D4 Endeavor diffractometer (Billerica, MA, USA). Radial scans of intensity versus scattering angle (2*θ*) were recorded at room temperature in the range of 2–40°(2*θ*) (step size = 0.02°(2*θ*), scanning rate = 4 s/step) with filtered CuKα radiation (λ = 1.54 A), an operating voltage of 40 kV, and a filament current of 40 mA.

Fourier infrared (FTIR) spectra of the fibers were recorded by a Jasco FT/IR-6200 (Madrid, Spain) equipped with an attenuated total reflection (ATR) accessory in the range of 400–4000 cm^−1^ in transmission mode.

Thermogravimetric analysis (TGA) was performed in a TG-STDA Mettler-Toledo analyzer, model TGA/STDA851e/LF/1600 (Columbus, OH, USA) to evaluate the influence of fibers on the thermal stability of PHBV. An initial mass of 10 mg of the materials was heated from 30 to 900 °C at 10 °C/min under nitrogen. The onset at 5% weight loss (*T*5%) and the maximum decomposition rate temperatures (*Tmax*) of fiber (f) and polymer (p) was determined from the weight loss and the derivative curves, respectively.

The rheological behavior of the materials was studied using a Discovery DHR-1 oscillatory rheometer (TA Instruments, New Castle, DE, USA) equipped with 25 mm diameter parallel plate geometry with a gap of 1.5 mm. Sample pellets were dried at 60 °C for 24 h before testing. The linear viscoelastic region was first determined by performing strain sweep tests at a fixed angular frequency of 1 Hz from 0.01 to 1000%. Further, the frequency sweep experiments were conducted at a fixed strain of 0.1% sweeping from 100 to 0.01 Hz at 180 °C. Storage modulus (*G*’), loss modulus (*G*”), and complex viscosity were analyzed.

### 2.4. Biodegradability Assessment

The disintegration study under thermophilic composting conditions was carried out according to the indications of the international standard ISO 20200 [34]. The compost was prepared by mixing 40% sawdust, 30% grounded rabbit food, 10% vegetal mature compost, 10% corn flour, 5% sugar, 4% corn oil, and 1% urea. The mixture was sieved through 5 mm and 2 mm sieves, and the water content was adjusted to 55 wt.% (pH 6.5, C/N ratio 28.9). Films were cut into pieces of 2.5 × 2.5 cm^2^, dried at 40 °C under vacuum for 24 h, and weighed. Films were individually inserted in a plastic net to be identified afterward. In triplicate, materials were buried in 1 kg of the compost (1.2% of each sample per container) and incubated at 58 °C for 45 days. Compost was periodically stirred to ensure aerobic conditions. To track the changes over time, 3 specimens per sample were extracted at different incubation times (10, 15, 21, 24, 27, 30, 34, 37, and 45 days), washed, dried under vacuum for 24 h, and weighed. The disintegration degree (*D*%) was determined using Equation (1):(1)$$D\;\left(\%\right)=\frac{m_{i}-m_{t}}{m_{i}}\times 100$$
where *mi* is the initial dry mass of the piece and *mt* is the dry mass of the same piece at a specific incubation time.

Surface/micro-structure analysis by means of SEM observations, changes in molecular weight of the polymer, and microbiological characterization were performed on the film pieces to track the progression of disintegration.

The molecular weight (*Mw*) of the PHBV at different composting times was calculated from intrinsic viscosity using the Mark-Houwink-Sakurada equation. The diluted solutions procedure followed to determine the intrinsic viscosity is detailed in the Appendix A [35,36,37].

The microbiological study was conducted by the quantification of microbial cells on films and compost after 10, 21, 30, and 37 days of disintegration. Each film specimen was placed in a sterile tube containing 20 mL of phosphate-buffered saline (PBS) and vigorously shaken by vortexing for 1 min to achieve the maximum detachment of microorganisms. The same procedure was followed with 0.2 g compost. Microbial suspensions were serially diluted by duplicate and spread on the corresponding media. Actinomycete isolation agar (AIA) with glycerol (5%, *v*/*v*) was employed for actinomycetes quantification, potato dextrose agar (PDA) for molds and yeasts, and plate count agar (PCA) for aerobic bacteria. The media were supplemented either with streptomycin sulfate salt (0.05 g/L) or cycloheximide (0.01 g/L) to suppress bacteria and mold growth, respectively. Plates were incubated at 30 °C for 7 days in case of actinomycetes, molds and yeasts, and 4 days for mesophilic bacteria. In parallel, some PCA plates were incubated at 58 °C for thermophilic bacteria. Results were expressed as a log of colony-forming units (*CFU*) per gram of material. To determine statistical significance, microbial count data were subjected to analysis of variance (ANOVA) using Statgraphics Centurion XVI version 16.1.17 (Manugistics Corp. Rockville, MD, USA). Significant differences were determined using the least significant difference test (*p* < 0.05).

The biodegradation (or mineralization) test under industrial composting conditions was carried out following the international standard ISO 14855-1 [38]. First, pellets of the samples were ground and sieved through a 250 µm mesh to equalize their particle size. The three compositions (PHBV, PHBV/TC and PHBV/WF), a positive reference (cellulose TC40), and a blank were tested in triplicate: 15 g of mature vegetal compost and 2.5 g of the powdered test material in each bioreactor on a dry basis. The moisture of the mixture was adjusted to 50%. Airtight bioreactors were incubated at 58 °C for 90 days. Carbon dioxide evolved by microorganisms’ action was directly measured by a G110 IR analyzer (Fonotest, Madrid, Spain). Aerobic conditions were guaranteed during testing. The biodegradation degree (*B*%) was calculated using Equation (2):(2)$$B\;\left(\%\right)=\frac{CO_{2}\left(t\right)-CO_{2}\;\left(b\right)}{ThCO_{2}}\times 100$$
where *CO_2_ (t)* is the accumulated carbon dioxide at a specific time, *CO_2_ (b)* is the average accumulated carbon dioxide of the blank at the same time, and *ThCO_2_* represents the total theoretical carbon dioxide calculated from the total organic carbon (TOC) and the mass of each sample. TOC of the materials (in g C/g sample) was determined by elemental analysis, obtaining 0.53–0.55 for PHBV and both composites and 0.44 for pure cellulose TC40.

## 3. Results and Discussion

### 3.1. Characterization of Fibers and Composites

Figure 1 gathers the results of the morphological and chemical characterization of PHBV, fibers, and composites. TC90 (Figure 1b) presents a bar-shaped morphology with a smooth surface. The lengths of the fibrils varied from 20 to 120 µm with a high aspect ratio of 10. By contrast, WF (Figure 1d) shows a heterogeneous size distribution varying from 400 to 1400 µm with a lower average aspect ratio of 4. Morphologically, WF is composed of an irregular particle mixture of bigger rectangular-shaped chips and smaller and thinner sticks. Its surface presents high roughness with lengthwise cracks. At higher magnification, pores associated with the dry wood cells are observed in bigger particles.

Woodflour presents higher percentages of both moisture and ash (7.6% moist., 0.39% ash) than cellulose TC90 (5.7% moist., 0.26% ash) due to higher contents of hemicellulose and lignin, respectively. Hemicellulose is the main responsible for water absorption by intermolecular hydrogen bonds with hydroxyl groups (–OH), although cellulose and lignin also contribute [39,40]. Regarding lignin, during pyrolysis, fragments of degraded hemicellulose and cellulose connect with the lignin structure. This leads to the stabilization of the lignin, increasing the residue [39]. Characterization of fibers in terms of crystallinity (WAXS), thermal stability (TGA), and chemical structure (FTIR) can be found in the Appendix A) [16,40,41,42,43,44,45,46,47,48].

Regarding the morphology of polymers and composites, PHBV monophasic surface (Figure 1a) is only interrupted by laminar inclusions of boron nitride used as a nucleating agent in the commercial grade. At low magnification (upper micrographs of Figure 1c,e), homogeneous dispersion of both fibers in the polymer matrix is observed. At high magnification, good fiber-matrix adhesion and no clear pull-out effect are detected. It could be explained by the interaction of hydroxyl groups (–OH) of cellulose with the carbonyl groups of the polyester [49]. In the case of WF, mechanical adhesion of the polymer to the fiber was found, leading to an effective attachment. This can be deduced by the cohesive breakage of WF particles. The roughness and porosity of fibers may allow the molten polymer to penetrate the structure during melt blending, and once solid, the interphase remains mechanically anchored [5].

The thermal stability of each component of the composites is affected by the other (see curves and parameters obtained in Appendix A and Table 1). Neat PHBV showed a single weight loss step centered at 297 °C. PHBV degradation occurs by a random chain scission reaction leading to a drastic decrease in molecular weight [50,51]. This reaction takes place via cis-elimination in a six-membered ring and releasing crotonic acid that may autocatalyze the decomposition of the polyester [52].

In composites, processing polyesters in the presence of solid particles generates higher shear stresses, causing local overheating due to friction [53] and, thus, promoting degradation. Pure cellulose barely affected *T*5%, but the presence of woodflour decreased it by 13 °C with respect to neat PHBV. Similarly, *Tmax.p* of PHBV was slightly reduced in TC composite while in WF composite, *Tmax.p* shifted 7 °C towards lower temperatures. Regarding the peaks of the fibers, both TC and WF were affected by compounding with PHBV, *Tmax.f* decreasing 7 °C. Pyrolysis of each component generates different degradation products that can react with the other ones. According to Fraga et al. [54], crotonic acid produced in PHBV degradation may promote the hydrolysis of cellulose, while in the case of WF composite, the acetic acid formed in the degradation of hemicellulose may act as an acid catalyst and initiate the earlier degradation of PHBV. Therefore, the additional presence of hemicellulose in PHBV/WF reduced the thermal stability of the polymer matrix in a significative way rather than in PHBV/TC.

Polymer–filler and filler–filler interactions are two of the most important factors related to chain mobility and particle dispersion, respectively, influencing the rheological behavior of composites [55]. In this study, the values of storage modulus (*G*’), loss modulus (*G*”), and complex viscosity were found to be PHBV/TC > PHBV/WF > PHBV throughout the full range of frequency (Figure 2). The increase in both magnitudes with the addition of cellulosic fillers is more prominent at low frequencies. In particular, for *G*’, both composites presented an abrupt gap with respect to PHBV, indicating a more elastic or solid-like behavior due to the rigid character of fibers. In general, fibers act as physical cross-linking points that hinder the relaxation of polymer chains, thus increasing the storage modulus [56]. The fiber–polymer interactions were found to be higher for the composite having pure cellulose than those of the woodflour. This effect cannot be only attributed to the presence of lignin [57] but also can be caused by the larger particle size of WF.

Relevant differences in complex viscosity have been found (Figure 2c): the arrows in the figure show that pure PHBV presented a decreasing curve for frequencies below 0.1 Hz, PHBV/TC exhibited an increasing curve, while PHBV/WF showed a plateau. Since the viscous response is predominant in that region after long-time measurements, the molecular weight of PHBV may be thermally reduced, showing lower viscosities. As previously described, the presence of the fibers affects the rheology of the polymer: the bigger the particle-specific surface and fiber-matrix interactions, the higher the effect on the polymer chain mobility. Hence, for the composite samples, polymer chain movements were refrained, and chain relaxation was hindered by the fibers, thus leading to higher viscosities that were able to counteract the natural viscosity decrease associated with the polyester thermal degradation. This effect was found to be higher for TC than WF composites due to lower particle size and higher ability of pure cellulose to form hydrogen bonds.

### 3.2. Biodegradation Assessment

Degradation of polyhydroxyalkanoates occurs by two overlapped steps: heterogeneous enzymatic hydrolysis of polymer chains into small soluble molecules and oligomers (disintegration) and subsequent metabolizing of this low molecular weight species by bacteria into CO_2_ and H_2_O in aerobic conditions (known as biodegradation, bioassimilation, or mineralization) [58].

Disintegration (*D*%) was determined by monitoring weight loss as a function of composting time, which results are plotted in Figure 3a. Although color change occurred from the first extraction on day 10, no remarkable weight loss was appreciated until day 21 for any sample (induction time). From this point, the disintegration rate abruptly increased, while a roughening of the surface of all the tested samples was clearly observed. PHBV/TC showed the highest disintegration percentages in the period of 15–25 days.

During the induction time, it is reported that a biofilm is established on the sample surface, and the extracellular enzymes of the microorganisms begin the hydrolysis of the polymer [5,49,59]. The degradation of PHBV takes place from the surface into the bulk, as the extracellular enzymes responsible for the hydrolysis of the ester groups cannot enter much deeper into the polymer matrix due to its high crystallinity and low hygroscopicity [58,60,61]. Furthermore, tightly packed polymer chains in crystalline regions do not allow water to penetrate the structure, moisture being a key factor for microbial survival and for supporting the hydrolytic process of degradation [29]. Apparently, considering the average disintegration values, it can be deduced that pure cellulose (TC) accelerates the process while the presence of the lignocellulosic filler hinders it.

Comparing the observations made by Avella et al. [23] and Sánchez-Safont et al. [5] in wheat straw and almond shell, not all lignocellulosic fillers seem to have the same effect on biodegradability in composting conditions. Beyond the chemical composition and the sole presence of lignin, the size and the structure of the particles of each lignocellulosic filler may influence the degradation process. The higher particle size of WF could reduce the surface of the polymer exposed as well as the presence of lignin all over the surface of the filler could increase hydrophobicity. In the case of TC, the small particle size of easier degradable and hygroscopic cellulose could promote degradation by disrupting the polymer surface, creating more accessibility points with enough moisture. Higher adhesion could facilitate the connection and degradation of both polymer and cellulose, creating favorable paths for microorganisms. However, the differences among samples (Figure 3a) were not statistically significant (*p* > 0.05). Moreover and regardless of the rate, all samples reached the same disintegration level of 89–90% at 37 days and completely disappeared at 45 days.

For the sake of comparison, relative molecular weight (*Mw*) was considered in order to minimize the differences between the starting *Mw* of each material due to the additional thermal treatment of composites during blending. The percental decrease in the *Mw* is plotted in Figure 3b. As expected, the decrease in *Mw* fully correlated with disintegration results, PHBV/WF being the slowest material. An initial slow trend was detected for the first 21 days, in which *Mw* remained over 75% of the original one. After this initial time, *Mw* dropped sharply, which agrees with a superficial biodegradation mechanism. During this initial time, the microorganisms attach to the surface, and only enzymatic degradation is performed, which causes a slow reduction in *Mw* [62]. The highest reduction was found from 21 days of composting when around 30% of the *Mw* of the three materials was lost in only 4 days. After degrading the surface, the microbes could move towards the inner core of the materials, rapidly decreasing their *Mw* [62]. *Mw* measurements beyond 30 days could not be performed due to the experimental limitations of the method.

Micrographs of recovered specimens displayed in Figure 4 revealed that during the induction period, colonization of the surface by bacteria and fungi took place, as well as a slight heterogeneous surface erosion. After 21 days, when roughening is evident, SEM micrographs showed the uncovering of both fibers resulting from preferential disintegration of the polymer rather than of the fillers, those remaining apparently unaltered. This contrasts with the holes and colonization by fungi hyphae found by Chan et al. [24] in PHBV/wood composites. The preferential consumption of the polymer may lead to the detachment and eventual release of the fibers, thus contributing to a higher weight loss rate observed for the PHBV/TC sample. However, the bigger size of the WF particles with respect to the TC will require larger disintegration to release the fillers.

Biodegradation is known to take place at a higher rate in amorphous structures than in crystalline polymers [63]. The accessibility of enzymes to the amorphous part of the polymer may be easier due to its higher chain mobility [61]. Moreover, crystallites do not allow water to enter, and therefore, the enzymes cannot degrade them as fast as the amorphous fraction. As observed in Figure 4, a clearly eroded and porous amorphous phase was still present on the surface at 27 days. At 34 days, when most of the amorphous fraction has already been biodegraded, the crystalline phase is revealed. PHBV/WF showed a star-shaped geometry identified as the spherulites of the polymer [61], while the crystalline structure of PHBV/TC appeared under a holed geometry. These holes could be left by bacterium colonies without any selection between the amorphous and crystalline parts [63].

Although the degradation of fibers was not detected in the disintegration test, it is universally accepted that lignocellulosic materials are biodegradable. Measurements of the carbon dioxide evolved were used to confirm if all carbon present is fully mineralized by microorganisms (Figure 5). In less than 60 days, all materials studied, as well as the cellulose used as a reference, reached a plateau indicating the end of the process. The ultimate aerobic biodegradation accomplished by reference, PHBV, PHBV/TC and PHBV/WF was 97%, 100%, 99% and 97%, respectively.

However, there are differences in the speed of the materials analyzed that can be attributed to their different chemical composition. PHBV showed the highest biodegradation rate, followed by PHBV/TC, which presented a slightly lower rate beyond day 15. The disparity between both curves is especially higher once 80–85% of biodegradation is achieved. Although cellulose is considered a fast and fully biodegradable biopolymer, being used as a standard for this kind of experiment [64], it exhibited a lower biodegradation rate than PHBV. Compared to the amorphous part of PHBV, the strong-packed crystalline structure of cellulose required prolonged time for the enzymatic attack [65]. The biodegradation rate of PHBV/WF drastically slowed down in comparison to neat PHBV and PHBV/TC after the first week of incubation. The components of woodflour have different degradation rates: hemicellulose > cellulose » lignin [64]. Bonding between lignin and hemicellulose supposes a barrier for microorganisms to access cellulose [66], decreasing the overall degradation rate of the composite. Additionally, slow-degrading particles of woodflour dispersed in the polymer matrix may limit the accessibility of water and microorganisms to PHBV [65].

The degradation of lignin is a complex process in which many microorganisms and enzymes are involved. Among others, white-rot fungi are considered the most efficient lignin degraders [67]. Lignin is able to degrade, but the process takes years or decades [64]. Taking into account the percentage of lignin in softwood (25–35%) along with the fraction of filler (15%) added to PHBV, around 4–5% of lignin is present in the composite. Given the ultimate biodegradation degree of PHBV/WF as 97.1 ± 0.3%, biodegradation of lignin fraction could not be asseverated.

Regarding the criteria established by international standard ISO 17088 [68] for plastics or EN 13432 [69] specifically for packaging, a polymer can be considered industrially compostable under aerobic conditions when disintegration (*D*%) and biodegradation (*B*%) reach at least a 90% in a maximum of 3 and 6 months, respectively. All materials studied fulfilled these statements even in case lignin would have not degraded. Although bioplastics are not the only solution to plastic pollution, the positive results obtained at the laboratory scale establish the base for consideration of these PHBV/lignocellulose biocomposites to further (and necessary) field-scale biodegradability tests.

### 3.3. Microbial Dynamics over the Composting Process

Compost comprises a wide diversity of mesophilic and thermophilic microorganisms, including fungi and bacteria, which are the main factors responsible for biodegradation [70]. Thus, information about their abundance and population dynamics throughout the process might help to better understand how degradation occurs.

All the studied microbial groups in compost exhibited a significant effect (*p* < 0.05) of incubation time, as demonstrated in Figure 6. Among them, fungi (Figure 6a) were the most influenced, with no detection after 21 days. This observation is consistent with visual assessments that showed a whitish layer formed by fungal mycelia during the initial days of incubation which subsequently vanished. Furthermore, fungal counts were relatively low compared to other groups, which can be attributed to fungi’s reduced thermo-tolerance and greater activity in mesophilic processes [71,72]. The statistical analysis indicated a significant influence of the type of material on fungal count (*p* < 0.05), as shown in Figure 6a.

Similar observations were reported by Galitskaya et al. [71], who found minimal levels of fungi in the compost after 30 days of thermophilic incubation, and by Husárová et al. [73], who did not detect fungi in compost incubated at 58 °C. Karamanlioglu et al. [74] also pointed out a detrimental effect of temperature on fungi when characterizing fungal communities associated with the degradation of poly(lactic) acid (PLA) in compost. Likewise, no fungi were detected on samples for longer times, although in this case, their absence was delayed until day 37.

While fungi were present in compost and samples after 10 days, they were not visible in colonizing samples until day 21. SEM evidence of such colonization is shown in Figure 7d,e. According to SEM, fungi hyphae appeared in PHBV at 15 days (Figure 7a). The nature of the cellulose within the matrix was revealed to be of great influence. While WF seemed to hinder fungal development, TC favored it (Figure 7h), giving rise to the highest fungal counts at day 30 (Figure 6a). It is worth noting that PHBV/TC samples experienced the most weight loss during disintegration, while PHBV/WF samples showed the least. Fungi are believed to be instrumental in breaking down PHBV with and without lignocellulosic residues, particularly during the initial stages, due to the vast array of enzymes they possess, including extracellular ones [74,75].

Other relevant microorganisms in composting are actinomycetes, one of the largest taxa within bacteria that produce a great variety of enzymes and represent approximately a third of the total prokaryotic life in soils and compost [76,77]. Several studies have previously reported their ability to degrade biopolymers, including PLA and PHAs [73,78]. The highest load of actinomycetes in compost was found after 21 days of composting when their population increased by more than two logarithmic units (Figure 6b). The same trend was observed for mesophilic bacteria (Figure 6c). Actinomycetes and mesophilic bacteria are likely to require an initial adaptation period to the composting conditions before they begin to proliferate, reaching their peak development after three weeks. Subsequently, their populations decline in both cases, possibly due to competition with more prevalent species, such as thermophilic bacteria. Neither actinomycetes nor mesophilic bacteria were detected in none of the PHBV samples in the first 10 days, indicating that they would not still be able to colonize polymer surfaces. However, beyond 10 days, populations of these microorganisms gradually increased over time in all samples. By day 37, the highest counts were detected in all materials. In terms of the impact of lignocellulosic residues, there were no significant differences found between TC and WF materials for actinomycetes overall (Figure 6b, with similar group numbers). However, TC did result in slightly higher amounts of actinomycetes on all counts. Similarly, TC incorporation resulted in significantly higher (*p* < 0.05) loads of mesophilic bacteria compared to neat PHBV.

In relation to thermophilic bacteria, their load in compost at day 10 was much higher than that of actinomycetes and mesophilic bacteria (7.2 log *CFU*/g versus 5.2 and 4.7, respectively), probably because temperature conditions were more favorable to their growth. Their population kept increasing until day 30 and sharply declined by almost two logarithmic units by the end of the experiment. In samples, thermophilic bacteria behavior was more variable and, overall, were more abundant even in the first 10 days. After 21 and 30 days, count values decreased and then increased again, reaching their highest values at day 37. Contrary to that observed for mesophilic bacteria, there was no statistically significant influence (*p* > 0.05) of the incorporation of TC and WF.

The presence of various microbial groups in both compost and samples is likely to result in complex interactions among them. Even the specific characteristics of each material may influence their dynamics and the biofilm formed on their surface as well as the surrounding environment. This biofilm is expected to contain species able to initiate the biodegradation of high molecular weight polymers, such as true degraders, and also commensal microorganisms, which use oligomers, monomers, and products of degradation [79]. These interactions may also enable some microorganisms to grow or tolerate the high temperature that occurs during composting, even if it is not optimal [75]. Nonetheless, we are aware that the methodological approach used for the microbiological study has limitations, such as slow-growth fungi or non-culturable species; hence, the recovered microorganisms do not completely represent the actual community [72,74]. Further studies are needed and will be performed in future works to address these limitations.

## 4. Conclusions

The study investigated the effects of lignin on the performance and degradation of PHBV/fiber composites (85/15) prepared by melt blending using pure cellulose (TC) and woodflour (WF) as fillers. Thermal stability was reduced in woodflour composite due to the additional presence of hemicellulose, whose degradation products formed during pyrolysis catalyzed the degradation of PHBV. In terms of rheological behavior, the lower matrix-fiber interactions in PHBV/WF composite compared to PHBV/TC were evident in the lower values of modulus (*G*’) and complex viscosity. This could be attributed to the presence of lignin and the larger particle size of WF, leading to a lower contact surface. Compostability was assessed by monitoring mass loss and CO_2_ release. All materials achieved complete disintegration in 45 days, with PHBV/TC showing the fastest disintegration rate and PHBV/WF the lowest. This is in accordance with the microbial characterization results in which mesophilic bacterial and fungal counts were significantly higher for PHBV/TC rather than for PHBV/WF. Fungi and yeast happen to be key factors in the initial stages of the process. SEM analysis showed that after the colonization of the surface, PHBV was eroded layer by layer, starting from the amorphous phase and leading to the detachment of the fibers. Neither cellulose fibers nor woodflour disintegration could be confirmed by SEM. Biodegradation (mineralization of organic matter into CO_2_ and water) was achieved by all materials in less than 60 days. Differences in the biodegradation kinetics were attributed to the presence of lignin from woodflour, restraining the accessibility of fungal enzymes, bacteria, and water to the easier degradable PHBV and cellulose molecules. PHBV/WF (85/15) was found to be suitable for developing fully compostable biocomposites, but the thickness of the product, the particle size of the filler, and the amount of lignin should be considered for optimal biodegradation rates of the final products (thermoformable trays, single-use cutlery, etc.). Thus, future and complementary research on microbiological aspects at the molecular level of these composites may lead to new approaches, such as biostimulation or combined aerobic and anaerobic treatments, for improved bioplastic waste management technologies.

## Figures and Tables

**Figure 1 polymers-15-02481-f001:**
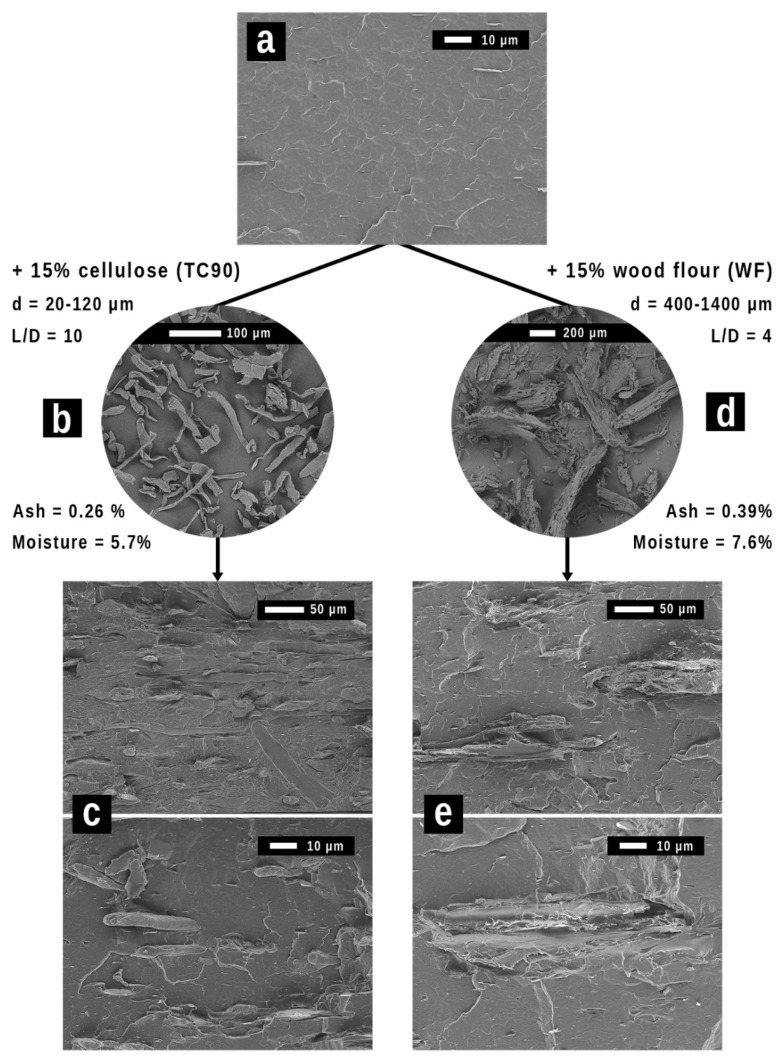
Micrographs of (**a**) neat PHBV, (**b**) cellulose, (**c**) 85/15 PHBV/cellulose composite, (**d**) woodflour, and (**e**) 85/15 PHBV/woodflour composite. Particle size, aspect ratio, moisture, and ash of fibers are gathered.

**Figure 2 polymers-15-02481-f002:**
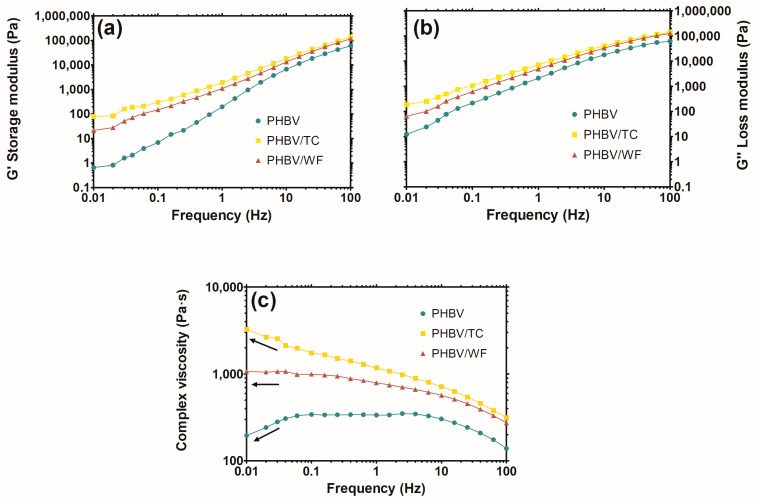
Frequency vs. storage modulus (**a**), loss modulus (**b**) and complex viscosity (**c**) of neat PHBV and fiber-composites.

**Figure 3 polymers-15-02481-f003:**
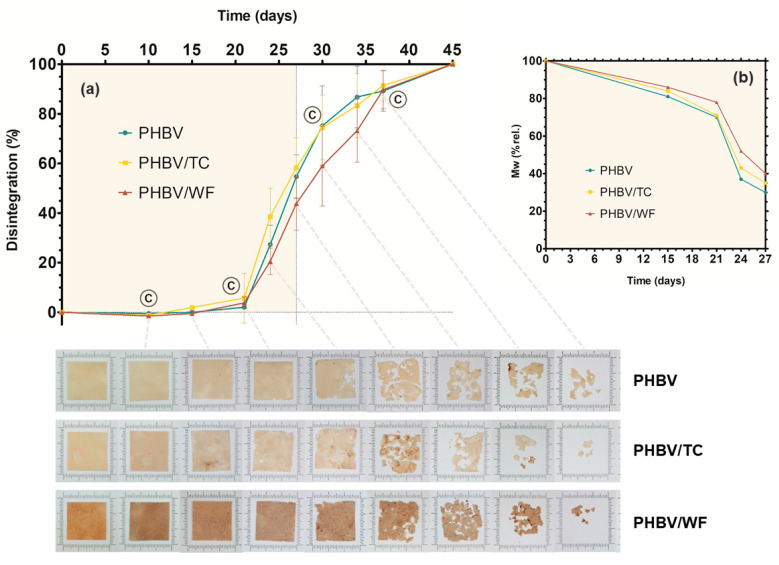
(**a**) Percentage of disintegration (weight loss) and pictures of the 25 × 25 mm pieces of PHBV, PHBV/TC and PHBV/WF over 45 days in thermophilic composting conditions. Error bars represent standard deviation. Letter C indicates the day when microbial counts were performed. (**b**) Relative *Mw* in the percentage of PHBV, PHBV/TC and PHBV/WF over 27 days in thermophilic composting conditions.

**Figure 4 polymers-15-02481-f004:**
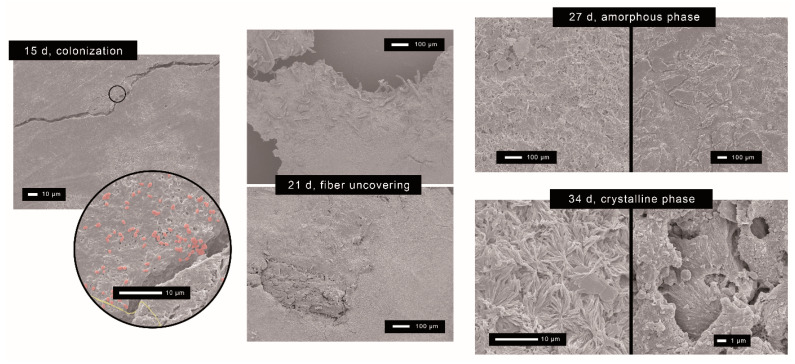
Representative micrographs of the composting process: 15 days, colonization of the surface; 21 days, releasing of fibers of pure cellulose (**up**) and woodflour (**down**); 27 days, degradation of the amorphous phase of PHBV/WF (**left**) and PHBV (**right**); 34 days, crystalline phase of PHBV/WF (**left**) and PHBV/TC (**right**).

**Figure 5 polymers-15-02481-f005:**
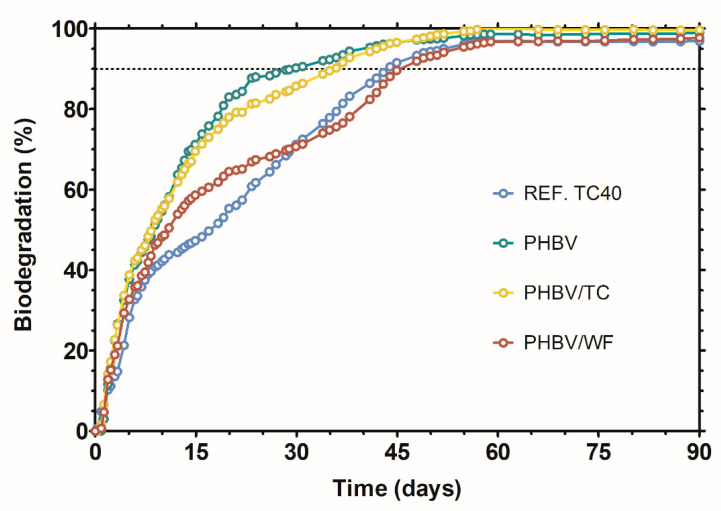
Biodegradation curve (*B*%) over time of neat PHBV and composites with purified cellulose (TC) and woodflour (WF) in composting conditions.

**Figure 6 polymers-15-02481-f006:**
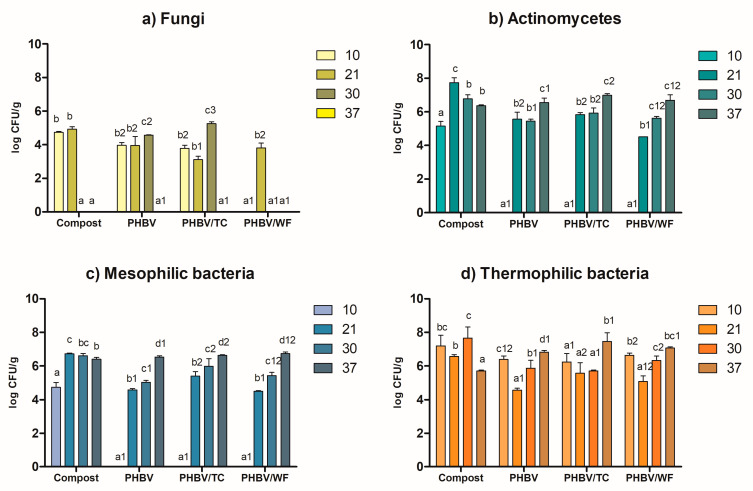
Microbial population and its evolution over time in compost and the different analyzed materials: (**a**) fungi; (**b**) actinomycetes; (**c**) mesophilic bacteria; (**d**) thermophilic bacteria. For each type of microorganism: the letters (a–d) above the bars represent significant differences (*p* < 0.05) in population for a particular material and different composting times; the numbers (1–3) indicate significant differences in the counts between different materials and the same composting time. For example, in Figure 6d, the bar for PHBV/TC at 30 days is represented by a1, meaning that the count is statistically similar to those of the same material at days 10 and 21 (represented by the letter a). However, PHBV/TC counts were similar to those of PHBV at 30 days (both represented by number 1) but different from PHBV/WF (represented by number 2).

**Figure 7 polymers-15-02481-f007:**
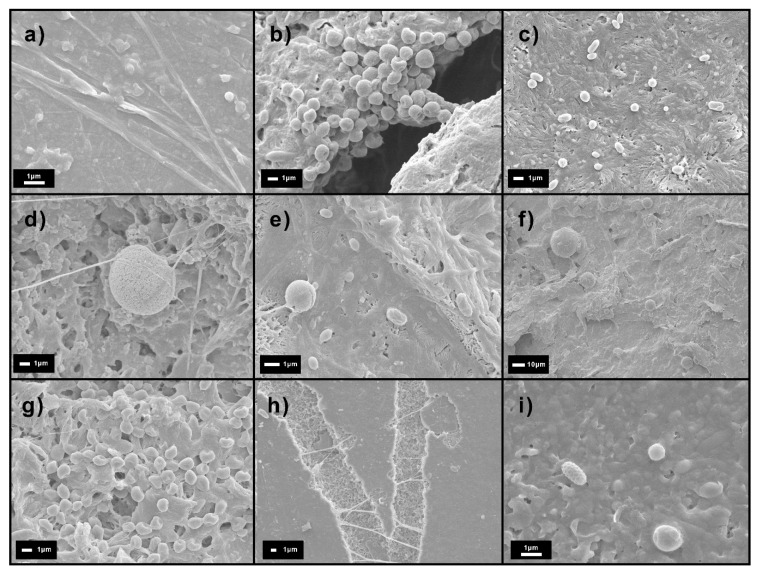
Different microorganisms found on the samples over composting time: hyphae (**a**) and bacteria colony (**b**) found on PHBV and PHBV/WF at 15 days; coccus and bacillus bacteria (**c**), spore surrounded by hyphae (**d**), and bacteria and fungi (**e**) found on PHBV, PHBV/TC and PHBV/WF respectively at 21 days; spores of different size (**f**) and coccus colony (**g**) found on PHBV/TC and PHBV/WF at 27 days; hyphae crossing biodegraded ways (**h**) and elliptical-rugose spore (**i**) found both on PHBV at 30 days.

**Table 1 polymers-15-02481-t001:** TGA parameters for fibers, neat PHBV and studied composites.

	*T*5% (°C)	*Tmax.p* (°C)	*Tmax.f* (°C)
Cellulose TC90	288	-	351
Woodflour WF	273	-	359
PHBV	277	297	-
PHBV + TC	276	294	344
PHBV + WF	264	290	352

## Data Availability

All data are contained within the article and the Appendix A or available upon email request from the authors.

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
