# Peer review of "Effect of the Presence of Lignin from Woodflour on the Compostability of PHA-Based Biocomposites: Disintegration, Biodegradation and Microbial Dynamics"

_polymers, 2023, doi:10.3390/polym15112481_

Round 1

Reviewer 1 Report

Comments to the Authors

This paper investigates the role of lignocellulosic fibers on the biodegradability of PHBV/wood chip composites under laboratory-scale composting conditions and explores their potential as realistic alternative biomaterials by studying will properties. In the course of the study, scanning electron microscopy (SEM), microbial enumeration and molecular weight loss were used to investigate the degradation process. The research includes not only traditional biodegradation studies, but also addresses microbial aspects and explores the mechanisms behind the bio recycling approach using biodegradable polymers in a large number of short-lived plastic applications. Clearly, this is a meaningful research work that provides a good reference for future research and for solving and dealing with the current degradation problems of plastic products. However, there are still some details that need to be improved. Therefore, the manuscript should be published after revision with the following detailed comments:

1. Pay attention to some formatting issues, such as the first letter of the keyword in the article needs to be capitalized and the way it needs to be adjusted.

2. The frequency vs storage modulus data show that about rheological behavior, the addition of fibers supposed, in general, a reinforcement of the material and increasing viscosity as polymer filler interactions were stronger.

But, why? Please give sufficient explanation and add some references.

3. The conclusion and abstract are too wordy, please rewrite these two parts.

5. There are some grammatical errors in the text, please correct them yourself. Please revise the layout and formatting of the article carefully; illustrations and notes that appear unaligned should be fixed.

Reviewer 2 Report

The manuscript entitled “Effect of the presence of lignin from woodflour on the compostability of PHA-based biocomposites: disintegration, bio- degradation and microbial dynamics” is an interesting work. However, there are some issues in the text that need to be addressed before publication. I suggest the following changes and improvements:

1.     Authors need to improve the abstract section and underscore the scientific value added to your manuscript in your abstract.

2.     What is more, plastics mixed with organics hinder the management of both residues. The sentence needs to rephrase and eliminate the grammatical issue.

3.     The current structure of the introduction is not well organized and long. The authors need to be improved. Additionally, the last part needs to be revised considering the main theme/objectives and findings of the study.

4.     The novelty of this work should be stated clearly in the introduction section.

5.     What are the current research gap and the significance of this work?

6.     Results are shown in Fig. 2a. Very short sentence.

7.     The author should provide a comparison-finding table of this study with other reported studies.

8.     The authors must be explaining their work's potential for use in practical applications.

Reviewer 3 Report

The submitted manuscript describes a study on the effect of lignin from woodflour on the compostability of PHA-based biocomposites. I think the submitted study is interesting and deserves to be published in the journal Polymers. I have only a few minor complaints about the manuscript.

·         Why are all the grade symbols underlined?

·         All variables should be italicised.

·         Lines 100 and 236 should not have indentation.

·         When references are written consecutively, they should be abbreviated with backslashes. I.e. [14-16] instead of [14,15,16].

·         I have not found an explanation for the characters in Fig. 4 (bc, a12, etc.).

·         The references are not described according to the Polymers style.

Reviewer 4 Report

The research is interesting but should be completely rewritten and re-reviewed. It's too interesting to waste such valuable material. The work should be based on physical and chemical methods and not on SEM. SEM is a method that allows you to prove everything and in this case it is not objective. Additionally, the microbiological part is unacceptable. It is based on culture methods of microorganisms, which are not used in such studies. I recommend reaching for molecular methods, it will enrich the work and open new research perspectives.

Round 2

Reviewer 1 Report

This manuscript can be accepted and pubblished on Polymers now.